# Evaluation of the Anticancer Potential of Crude, Irradiated *Cerastes cerastes* Snake Venom and Propolis Ethanolic Extract & Related Biological Alterations

**DOI:** 10.3390/molecules26227057

**Published:** 2021-11-22

**Authors:** Mostafa I. Abdelglil, Sanaa O. Abdallah, Mohamed A. El-Desouky, Mohammad Y. Alfaifi, Serag Eldin I. Elbehairi, Aly F. Mohamed

**Affiliations:** 1Faculty of Sciences, Cairo University, Giza 12613, Egypt; sanaa.osman.aa@gmail.com; 2Biology Department, Faculty of Science, King Khalid University, Abha 9004, Saudi Arabia; alfaifi@kku.edu.sa; 3Cell Culture Lab, Egyptian Organization for Biological Products and Vaccines (VACSERA Holding Company), 51 Wezaret El-Zeraa St., Agouza, Giza 12654, Egypt; 4The International Center for Training & Advanced Researches (ICTAR–Egypt), Cairo 11647, Egypt; fahmy.aly@gmail.com

**Keywords:** PEE, IRR V, A549, PC3, apoptotic, IC_50_, CV, P53, Casp-3, Bcl-2, ROS, MDA, and GR

## Abstract

We aimed to evaluate the anticancer potential of crude venom (CV), γ irradiated *Certastes cerastes* venom (IRRV), and propolis ethanolic extract (PEE). IRRV showed a higher toxicity than CV, while CV-PEE showed higher toxicity than IRRV and CV against lung [A549] and prostate [PC3] cancer cells. Toxicity to [A549] and [PC3] cells was concentration and cell type dependent. In comparison to controls, apoptotic genes showed a significant upregulation of P53 and Casp-3 and a downregulation of Bcl-2. Also, induced elevated DNA accumulation in the [S] phase post PC3 cell treatment with IRRV and CV, as well as a significant DNA accumulation at G2/M phase after IRRV treatment of A549 cells. In contrast, PC3 cells showed a negligible cellular DNA accumulation after PEE treatment. Glutathione reductase [GR] was reduced in case of PC3 and A549 cell treated with IRRV, CV, and PEE compared with its values in untreated cell control. The Malondialdehyde [MDA] values in both cells recorded a significant elevation post IRRV treatment compared to the rest of the treatment regimen and untreated cell control. Similarly, IRRV and CV-PEE mix showed obviously higher reactive oxygen species [ROS] values than PC3 and A549 cell treatments with CV and PEE.

## 1. Introduction

Bioactive proteins and peptides such as phospholipase A2 (PLA-2), L-aminoacid oxidase (LAAO), and disintegrin are the main constituent of snake venoms which have shown diverse biochemical activities [1]. Cancer is related to rapid and uncontrolled cell proliferation. It has motility, invasion, angiogenesis, and metastatic features [2]. Cytochrome-C is released from mitochondria due to induced apoptosis as carried out by chemotherapeutic agents [3]. Integrins are a variety of snake venom proteins. Disintegrins are one of these proteins, representing a class of cysteine-rich non-enzymatic proteins available in the venom of many snake species. The proteins detected in venom could contribute to the creation of novel cancer therapies [4]. Due to the wide range of pharmacology connected with snake venoms, they have become appealing sources for research into possibly unique alternative therapies, since various medications have been developed and are currently on the market [1]. In vitro venom detoxification methods such as ionizing X-irradiation, γ irradiation, UV light, controlled iodination, marine algae, and leaf extracts have all been tested, but none has been successful in humans [5]. Honey bees collect propolis, a resinous substance. Bee propolis has biological applications such as antimicrobial, antioxidant, anti-inflammatory, and dioxin toxicity reduction [6]. ACTX-8 protein has a cytotoxic effect on a number of cancer cell types in vitro. ACTX-8 can stimulate cell death in a time and concentration dependent manner. ACTX-8-treated cells, cytochrome-C introduced into the cytoplasm, and the dispersion of mitochondrial membrane potential “MMP” were both observed, indicating that the mitochondrial pathway was implicated in ACTX-8-induced cell death and that the ratio of pro- apoptotic/anti-apoptotic Bcl-2 family member expression levels is still constant by using ACTX-8. On the other hand, bad and Bax were both translocated to the mitochondria from the cytosol, and immuno precipitation revealed that in mitochondria, Bak and Bcl-xL dissociation first occurred, followed by Bcl-xL and Bad binding [7]. Propolis, a natural honey bee product, has been used in many studies for its antibacterial, antifungal, antiviral,hepatoprotective, and immunostimulatory activity, as it contains Flavonoids, cinnamic acid derivatives, steroids, amino acids, and vitamins such as B1, B2, E, and C belong to important components of propolis. Also, the active compounds derived from Propolis were demonstrated in several publications to contain high content of caffeic acid, phenethyl ester (CAPE) and chrysin (5,7-dihydroxyflavone) those were investigated with regards to antitumor activities [8]. The use of such apoptosis enhancers as key components in cancer treatment results in a decrease in the number of cancer cells and a reduction in the size of tumors, and is considered an indicator of apoptosis inducers in cancer treatment [9]. The goal of this study was to assess the anticancer efficacy of snake venom, both crude and γ-irradiated, as well as a propolis ethanolic extract. We investigated the anticancer activity of the tested products in lung and prostate cancers in terms of apoptosis-related genes, oxidative stress, and cell cycle profile.

## 2. Materials & Methods

### 2.1. Source of Venom

The lab animal section of VACSERA, Egypt, provided lyophilized *Cerastes cerastes* snake venom. *Cerastes cerastes* snakes were milked to collect the venom. The venom was vacuum-sealed and stored at 4 degrees Celsius until needed.

### 2.2. Irradiation of Crude Venom

*Cerastes cerastes* snake venom was irradiated in The National Center for Research & Radiation Technology, Cairo, Egypt, using a Co-60 γ cell (GE 4000A) produced by the Canadian Atomic Energy Commission. The radiation dose rate was 1.26 Gy/s at the time of experiment. The dose was selected as it gets rid of venom toxicity while keeping immunogenicity [10].

### 2.3. Crude Propolis

Commercially available propolis was purchased from the Faculty of Agriculture, Cairo University Giza.

### 2.4. Preparation of Ethanolic Extractof Propolis (PEE)

At room temperature, maceration with unintentional trembling was used to extract finely ground propolis in a ratio of 20 g of bee propolis added to 100 mL of solvent. The extracts were collected and filtered after maceration for 7, 10, 20, and 30 days, either in the presence or absence of light. As solvents, 100 percent pure ethanol (Merck-Germany) was used, as well as distilled water diluted to 70 percent, 50 percent, and 30 percent (*v*/*v*). Commercial grain alcohol (96° GL) and pure distilled water were also used [11].

### 2.5. Evaluation of Total Proteins of Cerastes CerastesCV and IRRV

The research was carried out at Faculty of Agriculture-Cairo University Research Park “CURP”. Using the STAT LAB SZZL0148, ver. 5.5 spectrum, calorimetric analysis was performed (PEE–CV–IRRV–CV-PEE mix).

### 2.6. Cancer Cell Line

VACSERA’s tissue culture department provided [A549] and [PC3]. The cells were cultured at 37 °C in an incubator humidified with 5% CO_2_ in a medium of L-glutamine-amino acid RPMI 1640 (Lonza, Verviers, Belgium), enriched with 10% heat-inactivated (FBS) (Jouan, France). The manufacturer’s instructions were followed for maintaining the cells.

### 2.7. Cell Viability In Vitro (MTT Assay)

The effect of CV, IRRV, PEE, and CV-PEE mix on A549 & PC3 cell viability was evaluated using MTT assay, where both cell lines were grown in 96-well plates (Costar, Switzerland) as 10^4^ cells/well for 24 h in a humidified incubator at 5% CO_2_ atmosphere. The growth medium 100 µL/well replaced by serum-free medium containing two-fold serially diluted CV, IRRV, PEE, and CV-PEE mix. After one day of treatment, the treatment media was decanted and the cells were washed using phosphate buffer saline (PBS) at 250 L/well in triplicate, followed by 50 µL/well of MTT solution (0.5 mg/mL in PBS) dispensed to the entire plates and incubation for 3 to 4 h at 37 °C. After carefully removing the supernatants, the formazan crystals were solubilized with DMSO (50 µL/well). The plates were lightly shacked for 15 min at 37 °C, A microplate reader was used to measure absorbance at 570 nm (Bio Tek- Winooski- VT- United States of America). The viability percentage was calculated using:
Percent of Viability = (A_570_ of treated-cells/A_570_ of control-cells) × 100.

### 2.8. Cell Cycle Analysis

Cell cycle distribution was examined by measuring the DNA content of nuclei labelled with propidium iodide (PI). Venom and propolis treated A549 and PC3 cell lines were pelleted post treatment by cold centrifugation (Jouan Ki-21-France), washed with 1 mL of cold PBS, centrifuged, and fixed in 70% cold ethanol at +4 °C for 24 h. Subsequently, cells were washed twice and treated with RNase A (20 mg/mL) and PI (20 mg/mL), FITC conjugated Annexin-V according to the protocol described by the manufacturer for 30 min at 37 °C in the dark. Finally, cell cycle distribution analysis was performed using flow cytometry and the percentages of cells at G1, S, and G2/M phases were calculated by flow cytometry (Becton-Dickinson, San Jose, CA, USA).

### 2.9. Isolation of RNA and Synthesis of cDNA

Cancer cell lines namely A549 and PC3 were harvested post treatment with the IC_50_ values, and the RNA was collected from the treated and untreated cells by the RNeasy Mini Kit (Qiagen, Valencia, CA, USA) according to the protocol of the manufacturer. Nano Drop 2000 was used to determine the concentration and purity of the isolated RNA. (Thermo Fisher Scientific, Waltham, Massachusetts, United States of America). cDNA synthesis was carried out in 10 µL reaction mixture containing 1µL of RNA sample, 2 µL of 10× RT Buffer, 0.8 µL of 25 (100 mM) dNTP Mix, 2 µL of 10× RT random primers, 1 µL of MultiScribe Reverse Transcriptase, 1µL of RNase inhibitor, and 3.2 µL of nuclease-free water using the cDNA Reverse The following temperatures and timings were used in reverse transcription using an Applied Biosystems thermal cycler: step one is 10 min at 25 °C, step two is two hours at 37 °C, step three is five minutes at 85 °C, and step four is held at 4 °C.

### 2.10. Real Time PCR-Quantitative

Once cDNA generated from non-treated and treated cells, SYBR Green real time PCR was carried out quantitatively. SYBR Green PCR master mix and a Step One real-time PCR machine (Applied Biosystems–Thermo Fischer Scientific, Waltham, MA, USA) were used to measure the expression of target genes. Each amplification reaction of qPCR was done in 20 × 10^3^ mL reaction mixture containing 10 × 10^−3^ mL Power SYBR Green PCR master mix (2×), 2 × 10^−3^ mL cDNA sample (100 ng), 1 × 10^−3^ mL forward primer (10 μM), 1 × 10^−3^ mL reverse primer (10 μM), and 6 × 10^−3^ ml double-distilled H_2_O. The conditions of cycling began with 10 min at 95 degrees Celsius, then by 40 two-step cycles of 15 s at 95 degrees Celsius and 1 min at 60 degrees Celsius. Melting curve analysis revealed the amplification. The relative gene expression was estimated using the 2—ΔΔCT method, where ΔΔ CT values of each sample were estimated from CT values; ΔΔCT = (CT target gene—CT GAPDH)”treated sample—(CT target gene—CT GAPDH” non-treated sample”. The primers used in the experiments were as follows:

Bax: F, 5′—GCTGGACAT TGGACT TCCTC-3′ and R, 5′—TCAGCCCATCTTCTTCCAGA—3′;

Bcl-2: F, 5′—TGTGGATGACTGAGTACCTGAACC—3′ and R, 5′—CAGCCAGGAGAAATCAAACAGAG—3′;

GAPDH (housekeeping gene): F, 5′—CGTCTGCCCTATCAACTTTCG—3′ and R, 5′—CGTTTCTCAGGCTCCCTCT—3′. The primers were formed by LGC Biosearch Technologies “Novato- CA-United States of America”.

### 2.11. Biochemical Analysis

A549 and PC3 IC_50_ treated cells (adherent and floating) were extracted, washed twice with cold PBS, centrifuged at 15 × 10^2^ RPM for 15 min, and pelleted (Jouan-Ki22—France). Sonication in an ice path was used to get the cell lysate, which was then cold centrifuged at 4 × 10^3^× *g* for 15 min at 4 ℃. The assay was performed on the supernatant. The levels of MDA, ROS, and GR activity were measured in both treated and untreated cells by readymade kits from Biodiagnostic (Cairo, Egypt) and a Milton Roy Spectronic 21 D UV-Visible spectrophotometer, according to the manufacturer’s instructions (Missouri, TX, United States of America). The Bradford method was used to quantify total protein in test cells.

### 2.12. Spectrophotometric Analysis

Spectrophotometer Thermo Scientific Helios Gamma was used in the analysis at Faculty Agriculture-Cairo University Research Park “CURP”.

### 2.13. Statistical Analysis

Each experiment was held in triplicates. Graph Pad Prism7 was used to estimate the IC_50_ values (Graph-Pad Software, La Jolla, CA, USA). Using Student *t*-test unpaired one, *p* values were determined by comparing each treatment group to the control group. *p* < 0.05 was used to determine whether the differences were significant on the statistical base.

## 3. Results

### 3.1. Cytotoxicity

The cytotoxic activity of *Cerastes cerastes* CV, IRRV, sole PEE, and CV-PEE mix to A549 and PC3 cancer cells was assessed using MTT assay, which revealed that venom toxicity was concentration and cell type-dependent, as viability increased as long as concentration decreased, Figure 1a,b, and the IC_50_ of IRRV being significantly lower than CV and PEE, and CV-PEE mix with IC_50_ values in the order of 18 ± 1.26, 40 ± 3.20, 117 ± 5.85, and 119 ± 7.14 μgm/mL for PC3cell line respectively and 11 ± 0.66, 20 ± 1.80, 85 ± 5.10, and 92 ± 7.36 μgm/mL for A549 respectively. Also, CV was significantly effective against A549 cell line than the PC3 cell line [*p* < 0.05], but IRRV had a minimal impact [*p* < 0.05] on both cell lines, and PEE was clearly more effective against the A549 cell line than the PC3 cell line [*p* < 0.05]. The findings were calculated using the average of three different experiments Figure 2.

### 3.2. Cell Cycle

It was reported at the G2/M phase in the case of PC3 cell treatment with IRRV, CV but not in CV-PEE-mix recording (33.4% 18.85%, and 5.5%), respectively, related to its values in untreated cell control (8.06%). At the same time, DNA content in the “G0-G1” and [S] phase showed a negligible change [*p* > 0.05] compared with that of the untreated cell control. In the meantime, A549 cells treated in the same sequence showed a clearly elevated percent of DNA content post A549 cell treatment with IRRV and CV-PEE mix, but not in the case of A549 treatment with sole CV, Recording (32.27%, 29.96%, and 2.96%), respectively, and as previous, there was a negligible [*p* > 0.05] DNA content observed in the [S] and [G0-G1] phases. Concurrently, there was clearly elevated total apoptosis [*p* < 0.05] compared to apoptotic values detected in non-treated cell control, and the IRRV apoptotic percent of the PC3 cell line was clearly [*p* < 0.05] elevated than CV and CV-PEE mix (42.33%, 16.6%, and 18.56%respectively). While there was a negligible [*p* > 0.05] change in apoptotic percent in PC3 cells treated with CV and CV-PEE mix (16.6% and 18.56% respectively), Finally, PEE treated PC3 cells showed negligible [*p* > 0.05] elevated DNA accumulation (11.11%), (40.24%), and total apoptotic percent (11.24%) at the [S and G2-M] phases, and total apoptotic percent compared with that of untreated cell control (8.06%, 39.88%, and 1.78%). At the same time, it was reported that DNA content during the G2-M phase was obviously increased [*p* < 0.05] post A549 cell treatment with IRRV (32.27%) and CV-EPP mix treatment (29.96%) compared with a control value (12.38%), and the DNA content during the [S] phase showed that CV showed in Obviously elevated value[*p* > 0.05] post A549 cell treatment with CV and CV-PEE mix (42.89% and 44.32%) but not post-treatment with IRRV and CV-PEE mix treatment. Finally, the DNA content during the [G0-G1] phase was negligible [*p* > 0.05] changed in comparison to the value in untreated cell control Figure 3a–d.

### 3.3. Gene Expression

Regarding the apoptosis regulating genes, it was noticed that PC3 cell treatment with CV, IRRV, PEE, and CV-PEE Mix showed significantly up-regulation [*p* < 0.05] of pro-apoptotic gene P53 post PC3 cell treatment with CV, IRRV, and PEE but not post PC3 cell treatment with PEE [*p* > 0.05], similarly Casp-3 was obviously up-regulated [*p* < 0.05] post cell treatment with the differently formulated venom compared with P53 value of untreated cell control. On the other hand, the Bcl-2 gene as an anti-apoptotic gene was significantly down regulated post different treatment. Similarly, the P53 gene was significantly [*p* < 0.05] up-regulated post A549 cells treatment with different venom formulae compared with its value in untreated cell control, and the same profile was detected in case of monitoring of Casp-3 gene. It was obvious that IRRV showed the highest up-regulation of P53 and Casp-3 genes. Meanwhile, there was a significant [*p* < 0.05] downregulated Bcl-2 gene and the lowest value detected in the case of using IRRV. The effect of test formulae on gene expression showed that p53 in IRRV was obviously elevated (*p* < 0.05) than in the case of A549 cells and CV-PC3 treated cells[*p* < 0.05], Also, P53 gene was significantly elevated than in case of A549 cell treatment with CV-PEE mix. Casp-3 gene in both cell lines was obviously more elevated post cell treatment with IRRV than post cell treatment with CV-PEE mix, Figure 4A–J and Figure 5 a,b.

### 3.4. Biochemical Analysis

Concerning the oxidative stress induced in venom formulae treated PC3 cells, it was noticed that GR showed significantly reduced values (*p* < 0.05) compared with untreated cell control post cellular treatment with IRRV, CV, CV-PEE mix and PEE respectively. On the other hand, A459 treated cells showed significantly reduced values (*p* < 0.05) compared with untreated cell control post cellular treatment with IRRV, CV-PEE mix, CV, and PEE respectively. There was an insignificant difference in the GR values in treated A549 and PC3 cells. Oppositely, MDA values was not cell type-dependent and MDA values for PC3 cell line and A549 showed a significant elevation post IRRV, CV-PEE mix, CV and PEE treatment compared to the test formulae and untreated cell control, Figure 6a,b. Similarly, IRRV, CV-PEE mix, CV and PEE showed significantly elevated ROS values compared to the rest of both treated cell lines, Figure 6c,d.

### 3.5. Evaluation of Total Protein

Total protein content measured showed that the protein content was 5.9 mg/mL, 13 mg/mL, and 11.3 mg/mL in PEE, CV, and IRRV, respectively. Due to the effect of radiation, the number of proteins in CV decreased, ensuring the venom detoxification process via irradiation. The CV-PEE mix was found to contain negligible reduced protein content (11.9 mg/mL), which may be due to the mixing process and the role of ethanol extraction induced protein content change compared to the CV Figure 7.

### 3.6. UV-Spectrophotometric Analysis

Spectrophotometer Thermo Scientific Helios Gamma was used in the analysis at “CURP”, Faculty of Agriculture, Cairo University. PEE, CV-PEE mix, CV, and IRRV are the acronyms for PEE, CV-PEE Mix, CV, and IRRV Figure 8a–d.

## 4. Discussion

One of the most common causes of mortality for a number of people worldwide is cancer, and despite significant advances in science, there are several obstacles that need to be overcome to be able to accelerate the cure of cancer. As a result, research in oncology is seriously developing novel and effective drugs that can help patients cope with fewer side effects than usual [12]. Systemic chemotherapy, which is used to treat cancer, has a lot of negative side effects. For a long time, scientists have concentrated on tumor treatments that harm the tumor-bearing host and its immune system [13]. As a result, the creation of a new anti-cancer agent that can differentiate between normal and malignant cells represents a significant advancement in cancer treatment. Snake venom has been offered for use in folk medicine. *Cerastes cerastes* venom, either CV or IRRV, in combination with PEE, had a cytolytic effect on lung [A549] and prostate [PC3] cancer cells in a concentration and cell type-dependent manner according to the findings. [PC3] was more susceptible to treatment with either CV or IRRV venoms than [A549], as seen by the IC_50_ values. The MTT results, which were involved in the study, proved the venom’s anti-neoplastic power against cancer cells compared to normal cells of the breast. The poisonous components of venom vary by species. Snake venom is a basic biological product that comprises a set of compounds with medicinal uses. As previously stated, snake venom is primarily composed of peptides and proteins with specific chemical and biological uses; these proteins are lethal in humans, but they can be useful in treating arthritis, thrombosis, cancer, and a variety of other disorders [14].

Overall, snake venom has been shown in studies to have potential as a new anticancer drug. Anti-angiogenesis and apoptosis-inducing agents are among the anticancer actions of snake venom derivatives. Venom disintegrins, which include leucurigin, contortrostatin, obtustatin, adinbitor, and salmosin, are a class of tiny cysteine-rich proteins identified from diverse snake venoms with anti-angiogenesis activity. In mice, snake venom contained disintegrins, including Leucurogin, which had anticancer and anti-angiogenesis properties. Contortrostatin reduces angiogenesis in human breast cancer primary tumors in mice. Obtustatin inhibited tumor growth in a mouse model and inhibited angiogenesis. In vivo and in vitro, adinbitor suppresses ECV304 cell proliferation and angiogenesis caused by bFGF. Salmosin has already been proven in lung cell carcinoma mouse xenografts to decrease the proliferation of bovine capillary endothelial cells as well as the growth of both metastatic and solid tumours [15]. External signals, such as the binding of cell surface death receptors to ligands, as well as signals from within the cellsuch as genotoxic stress, can cause apoptosis. Cancer is characterized by dysregulation of the apoptotic cell death mechanism. Apoptosis modification is responsible for the formation of tumors and development as well as resistance to treatments [16]. Anticancer activity was proved via the expression of anti-apoptotic and pro-apoptotic genes. Similar to our results, there was elevated apoptosis in a concentration-dependent manner. Cancer cells lose their ability to perform apoptosis, resulting in uncontrolled growth. Thus, inducing apoptosis may be a useful technique for cancer treatment. It has also been shown that some snake venom enzymes can induce apoptosis in cells [17]. According to the available literature, *Cerastes cerastes* venom contains LAAO-. L-amino-acid oxidase/Phosphlypase A2 (PLA-2) might trigger apoptosis in [A549] and [PC3] cancer cells treated with CV, IRRV, PEE, and CV-PEEmix. Indian cobra snake (Najanaja) disintegrins, which displayed anti-cancer and apoptosis induction activities in MCF-7, HepG2, and A549 cancer cells, were found to form induced apoptosis [14]. Endothelial cells are specifically affected by VAP and VAP2, two metalloprotease/disintegrin family members that trigger apoptosis. In ECV304 cells, stejnitin also causes apoptosis [15]. In terms of apoptosis-related biochemical alterations, it was discovered that there was a considerable increase in “ROS and MDA”, as well as a decrease in “GSH”. In addition, superiority to IRRV was higher than CV and PEE application, which could be related to the formation of novel tiny proteins that caused stronger biological activity after being irradiated. These findings were consistent with [18,19], which reported that most chemotherapeutic drugs increased intracellular ROS levels and disrupted cancer cell redox equilibrium. The redox balance distribution in tumor cells induces apoptosis. GSH protects cells from ROS buildup and apoptosis since it is the most prevalent anti-oxidant in cells. A decrease in GSH concentration in the cell is a sign of apoptosis [18,19]. The biological toxicity of venom LAAO content could be due to the second activity of H_2_O_2_ released during the oxidation of substrate. Adding GSH or catalase to LAAOs results in a reduction of activity of these enzymes, which has been demonstrated. The discovery that hydrogen peroxide is an apoptosis mediator that acts directly on oxidative cell metabolism adds to the growing body of research that supports this concept [20]. GSH levels in venom-treated cancer cells were found to be considerably lower in the current investigation. The decrease in GSH and increase in ROS values in cells shifted the redox balance in favor of apoptosis, and data showed that venom-induced apoptosis, likely via the mitochondrial pathway, changed expression of BCL-2 family apoptosis regulatory proteins, increased BAX/BCL-2 ratio, decreased mitochondrial membrane potential, and freed cytochrome-C to the cytosol from the mitochondria. CV from *Najaoxiana* and cardiotoxin III (CTX III) extracted from *Najanajaatra* venom induce cancer cell death [18,21]. Treatment of [A549] cell lines with 30 L/mL fraction 21 resulted in 40% lactate dehydrogenase release, an increased activity of 3.5 fold in caspase-3, and an increased activity of 3.2 fold in caspase-9, indicating that 30 L/mL of this part causes necrosis as well as apoptosis in the cells. The permeability of the mitochondrial membrane is responsible for this occurrence. When a toxin damages the inner membrane of the mitochondrial and permeability transition holes (“PTP”) form on the membrane, the potential of the membrane is lost, resulting in death due to necrotic factors. Cytochrome-C is released as a result of outer mitochondrial membrane damage, causing what is called cell death, and higher toxin concentration increases mitochondrial outer membrane permeability (MOMP) [19]. In Hu02 cells, fraction 21 concentrations produced necrosis, whereas the same concentrations caused apoptosis in [A549] cancer cells. In comparison to the control group, fraction 21 at 10 g/mL produced amazing results, with approximately 63 percent of the cells dying in [A549] cancer cells. Furthermore, mechanism fraction 21 inhibited [A549] cancer cell proliferation by activating caspases, and the increased activity of caspase-3 and caspase-9 suggested that this fraction causes cytotoxicity in cancer cells via the apoptotic intrinsic pathway [21]. Among the protein fractions 21 with two molecular masses were SVMP (P-I), CRiSP, and kallikrein [22]. CRiSPs are cysteine-rich secretory proteins present in snake venom that operate as L-type Ca^2+^ and cyclic nucleotide-gated (CNG) channel blockers in some species [17,18]. While CRiSP is not likely to be the cause of the reported cytotoxic effects, kallikrein enzymes, a type of serine protease, are involved in digestion, blood coagulation, neurotransmission, and protein post-translational modifications, among other phenomena. Crotalase, a kallikrein-like enzyme derived from *Crotalusadamanteus* venom, was demonstrated to reduce B16 melanoma cell growth in vivo by defibrinogenating mice. The enzyme had an indirect cytostatic or cytotoxic effect on cancer cells in vitro [23]. 

The cytotoxic effect of *P. persicus* venom fractions on Hu02 and [A549] cells was studied, and some fractions induced a decline in cell viability in both cells, either cancerous or normal. In colorectal and breast cells treated with snake-venom, there was a considerable increase in ROS. When venom was applied at varying concentrations, the effect was more powerful and universal (i.e., concentration dependent). In snake venoms, the presence of LAAO, which leads to ROS formation, could explain the increased expression of ROS [24]. Furthermore, when compared to their respective controls, the overall number of apoptotic cells increased by up to 70%. Furthermore, ROS has been shown to cause DNA damage and affect treatment susceptibility in cancer [25,26]. The oxidative stress condition is related to the production of free radicals by LAAO [27]. In neuroblastoma cells, an increase in ROS caused by venom of snake therapy increased pro-apoptotic proteins such as Bax, and there was a down-regulation of anti-apoptotic Bcl-2 protein after venom treatment. A similar process has been found in hepatocellular cancer, where bringing down IL-8 and HIF-1 raises the cytochrome-C quantity, which in turn enhances fragmentation of DNA and apoptosis. There was a decline in interleukin-6 expression in both cell lines after treatment with venom, which is likewise directly linked to the rise in the level of apoptosis. During cancer, ROS and some pro-inflammatory cytokines are always amplified [18]. In addition, Interleukin-6 has been shown to protect stomach cancer cells against apoptosis triggered by H_2_O_2_ [28]. Reduced production of interleukin-6 in venom-treated cell lines coincides with an increase in the number of apoptotic cells, which is consistent with our study. Since they are DR-related proteins that indicate cell death, the link between apoptosis and the production of apoptosis regulating proteins by snake venom toxin, caspase-3, 8, 9, Bax, and cytochrome-C was investigated. The whole cell extract was Western blotted after being treated with snake venom toxin (0.1–1 μg/mL). Cleavage of caspase-3, including cleaved caspase-3, caspase-8, including cleaved caspase-8, and caspase-9, including cleaved caspase-9, was seen in HCT-116 and HT-29 colon cancer cells. The Bax/Bcl2 ration was clearly raised, and cytochrome-C was obviously elevated in cytoplasm extract [29]. Similarly, the anticancer effects of bee Propolis have been linked to flavonoid content, particularly gallic acid. Furthermore, at 24, 48, and 72 h, Propolis extracts produced using two different extraction methods reduced the development of [A549] and HeLa cancer cell lines in a dose-dependent manner. The anticancer potential of several bee propolis samples obtained from various regions was tested on both the non-aggressive breast cancer cell line MCF-7 and the aggressive cell lines SK-BR-3 and MDA-MB-231. WST1 analysis was used to assess the anti-carcinogenic uses of bee propolis samples on SK—BR-3, MCF—7, MDA—MB—231, and only chosen ones on MCF—10A and hPdLF, initially based on phenolic content. In terms of phenolic/flavonoid chemicals, Turkey-originated propolis was greater than the other Propolis samples after HPLC and LS-MS/MS analysis. Both nonaggressive and aggressive BCCL cells were considerably inhibited by Turkey propolis (*p* < 0.01). Galangin, caffeic acid, apigenin, quercetin, and ferulic acid, which were applied to the MCF-7 cell line to determine cytotoxic and apoptotic effects, were linked to the toxicity of Propolis content. Galangin, apigenin, caffeic acid, and quercetin all inhibited cell growth in the MCF-7 cell line at all time intervals, but ferulic acid was ineffective. All cell proliferation inhibitions were apoptotic in nature, according to the annexin V-PI assay [30].

## 5. Conclusions

From the presented data, it can be concluded that snake venom, either CV, IRRV, PEE, or CV-PEE mix, has anticancer potential, assured via up-regulation of proapoptotic genes in combination with anti-apoptotic gene downregulation. Oxidative stress confirmed cell death and proliferation as there was a significant increase in ROS, MDA, and depleted GSH with cell death at the G2/M phase with a high percentage of cell apoptosis.

## Figures and Tables

**Figure 1 molecules-26-07057-f001:**
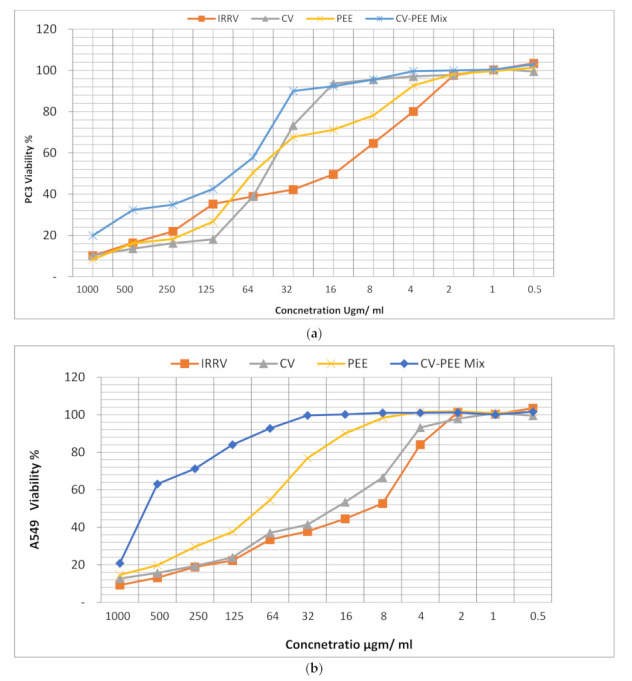
(**a**) Evaluation viability of IRRV, CV, PEE, and CV-PEE Mix against A549 cell lines post 24 h using MTT assay. (**b**) Evaluation viability of IRRV, CV, PEE, and CV-PEE Mix against PC3 cell lines post 24 h using MTT assay.

**Figure 2 molecules-26-07057-f002:**
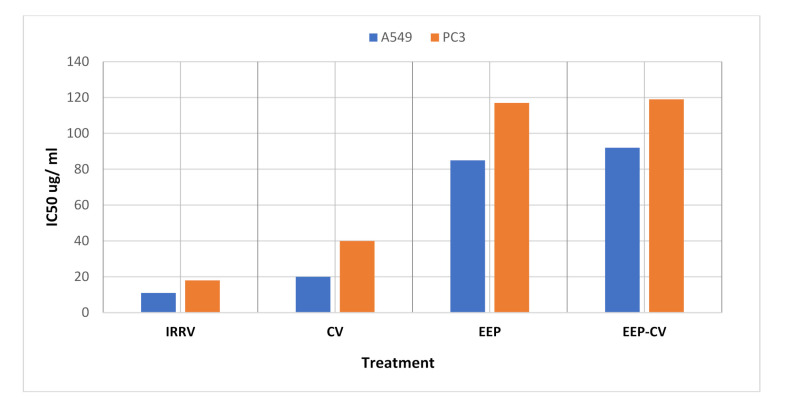
Evaluation of IC_50_ of CV, IRRV, and PEE against [PC3]&[A549]cell lines using MasterPLex2010.msisoftware.

**Figure 3 molecules-26-07057-f003:**
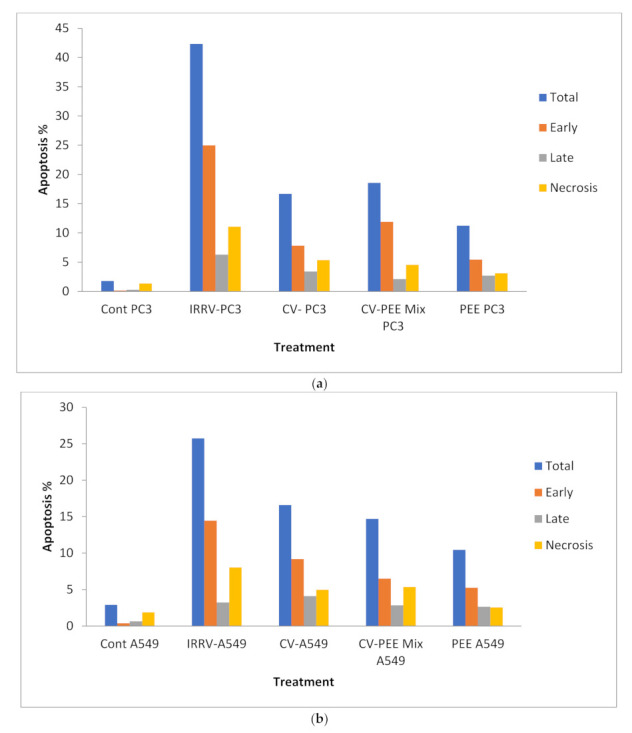
(**a**) Evaluation of apoptosis of PC3 cell lines with IRRV, CV, and CV-PEE Mix compared with cell cycle profile of untreated cell control. (**b**) Evaluation of apoptosis of A549 cell lines with IRRV, CV, and CV-PEE Mix compared with cell cycle profile of untreated cell control. (**c**) Evaluation of DNA content of PC3 cell lines with IRRV, CV, and CV-PEE Mix compared with cell cycle profile of untreated cell control. (**d**) Evaluation of DNA content of A549 cell lines with IRRV, CV, and CV-PEE Mix compared with cell cycle profile of untreated cell control.

**Figure 4 molecules-26-07057-f004:**
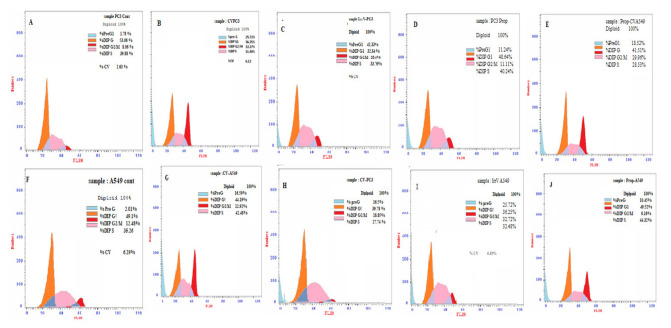
(**A**–**J**) Evaluation and DNA content and cellular apoptosis post [A549] and [PC3] cell treatment with PEE, CV, IRRV and PEE-CV mix using flow cytometry analysis.

**Figure 5 molecules-26-07057-f005:**
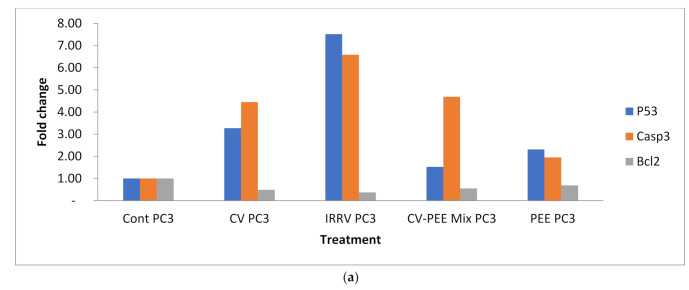
(**a**) Detection of PC3 cell line apoptosis-related genes in CV, IRRV, CV-PEE Mix, and PEE compared to untreated cell control using real-time PCR. (**b**)Detection of A549 cell line apoptosis-related genes in CV, IRRV, CV-PEE Mix, and PEE compared to untreated cell control using real-time PCR.

**Figure 6 molecules-26-07057-f006:**
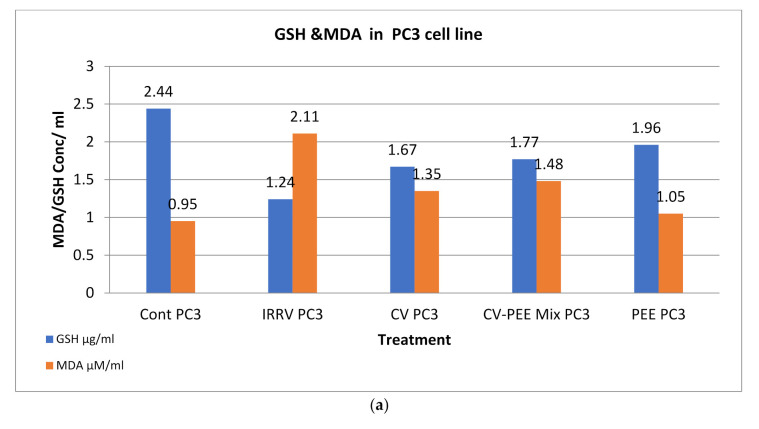
(**a**): Evaluation of glutathione reductase (GSH) and malondialdehyde (MDA) post [PC3] cell line treatment with IRRV, CV, CV-PEE Mix, and PEE. (**b**) Evaluation of glutathione reductase (GSH) and malondialdehyde (MDA) post [A549] cell line treatment with IRRV, CV, CV-PEE Mix, and PEE. (**c**) Evaluation of reactive oxygen species post [PC3] cell lines treated with IRRV, CV, CV-PEE Mix, and PEE. (**d**) Evaluation of reactive oxygen species post [A549] cell lines treated with IRRV, CV, CV-PEE Mix, and PEE.

**Figure 7 molecules-26-07057-f007:**
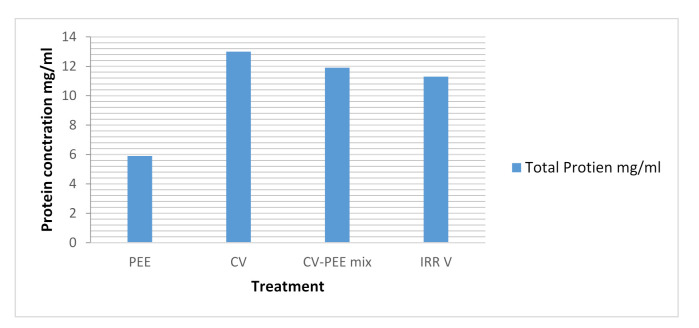
Evaluation of Total Protein of IRRV, CV-PEE Mix, CV and PEE.

**Figure 8 molecules-26-07057-f008:**
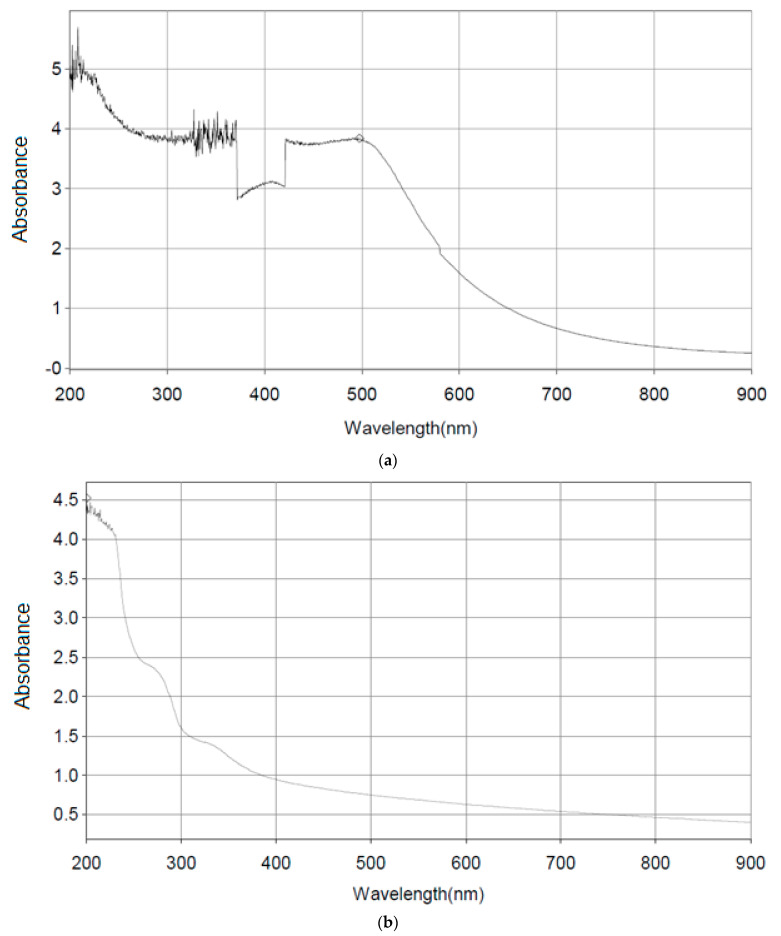
(**a**) Detection of PEE. UV Spectrum Starts at 200 nm and ends at 900 nm with absorbance starts at 5.046 nm and ends at 0.2252 nm. (**b**) Detection of CV. UV Spectrum Starts at 200 nm and ends at 900 nm with absorbance starts at 4.523 nm and ends at 0.401 nm. (**c**) Detection of CV-PEE mix UV Spectrum Starts at 200 nm and ends at 400 nm with absorbance starts at 5.222 nm and ends at 0.2252 nm. (**d**) Detection of IRRV. UV Spectrum Starts at 200 nm and ends at 400 nm with absorbance starts at 3.979 nm and ends at 0.291 nm.

## Data Availability

All data are available once requested from the corresponding authors.

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
