# Peer review of "Evaluation of the Anticancer Potential of Crude, Irradiated *Cerastes cerastes* Snake Venom and Propolis Ethanolic Extract & Related Biological Alterations"

_molecules, 2021, doi:10.3390/molecules26227057_

Round 1

Reviewer 1 Report

The aim of the present was to evaluate the bioactive potential of three products such as Certastes cerastes venom, in  its raw form, after controled irradiation and propolis extract. The topic of the study is not very new because there are some similar studies in the literature. I am curious what was the motivation when the two products were selected, a product of origin from a snake and a bee product? And why were they tested in parallel? Why did you choose to potentiate the effect of the venom with propolis? The introduction specifies certain data from the literature related to the composition of snake venom that can be correlated with the bioactive effect, not in the case of the bee product propolis. Consequently, I believe that it should be completed rigorously. I have several objections to materials and methods. In the case of the study, the agreement of the ethics commission is not specified.
Is it just the protein composition for the venom obtained? the rest of the compounds are not relevant for the biological effect, ie for the antitumor effect?
The chemical composition is not determined for propolis extract, just proteins, and are they enough for the bioactive effect? How can we correlate the biological effect if we do not have such data? Do you think that only the proteins in the composition are responsible for the biological effect?

For the used cells lines the origin is not described. The culture conditions are briefly described. To describe the microclimate, the authors use the following statement: the cells were cultured at 37 ° C in an incubator humidified with 5% CO2 in a medium. Yes the microclimate must be humidifed, but the humidity is not provided by CO2. For cytotoxic assessment of the selected substances, the authors did not use a positive control (a chemotherapeutic agent), and for propolis, ethanol as a control. And there is no absolute control. For the evaluation of the cell cycle the method is not described. The tested concentrations for substances are not described in materials and methods, how were the concentrations were established?

I consider that a restructuring, re-evaluation and completion is necessary for this article to be published.

Author Response

Dear Reviewer, Thank you for your comments

Reviewer 2 Report

Snake venoms are complex mixtures containing many different biologically active proteins and peptides. Bee propolis has biological applications such as antimicrobial, antioxidant, anti-inflammatory, and dioxin toxicity reduction. The goal of this study was to assess the anticancer efficacy of snake venom, both crude and γ-irradiated, as well as ethanol extracted propolis.

The abstract slightly exceeds the total of about 200 maximum words required. Generally, acronomies are not allowed in the abstract and some are not very clear, for example PEE-CV and CV-PEE.

The sections of the research manuscript are not arranged according to the directions of the journal, i.e introduction, results, discussion, materials and methods, conclusions.

References should also be reviewed. Figures should be improved, standard deviations are often not reported. In my opinion the work as it is is not adequate by Molecules standards.

Author Response

(The authors gave the same response as above.)

Reviewer 3 Report

This is very interesting paper describing different factors which may affect cancer cells. However some small problems are visible in the paper. 

  • The introduction and discussion part must be rewritten. The expalantion of mechasim of action of ACTX- 8 is preseneted in opposite manner. The release of Cyt-c from mitochondria is a result but not cause cytotoxic effect. the introduction generally explains the goal of experiment, however in discussion part many examples are unecessairy and could be moved to the Introduction part (like difference in venom concentration and activity in different snakes).

- Many of the drugs cannot be used due to the problem of application. How authors would like apply the venom into the patient and in what cases?

Author Response

(The authors gave the same response as above.)

Reviewer 4 Report

The manuscript entitled Evaluation of the anticancer potential of crude, irradiated Cerastes cerastes snake venom and Propolis Ethanolic extract & related biological alterations is very interesting and easy to read with exception of typing and spelling errors. Due to unclearness of figures (the Figures 1 and 2 are the some?, the figure 7 has no meannig, the abbreviations in figure 6 is not matched to the text in figure title, and double labels for graphs in figure 4)

The Figure 9 represents UV spectra, where absorbance is up to 5 units at wavelenght lower than 300 nm. At that point, the UV spectrometry is not appropriate method for detection.

As the theme is interesting and some experimental work was done, I suggest that the figures are rearranged and according to that the text is customized. In this form, the manuscript should not be published in this journal.

Author Response

(The authors gave the same response as above.)

Round 2

Reviewer 1 Report

The article has been partially improved, some aspects related to my evaluation have been clarified. I understand that in the literature there are data related to the chemical composition of propolis, but this composition depends on several factors such as plant sources, geographical area, season and obviously other factors. Regarding the effective antiproliferative effect, I cannot agree with the authors' answer. How did they determine the safe concentration for propolis and venom? My question referred to other issues. In any experiment in which the antiproliferative effect is evaluated, comparison with a control is imperative.

Author Response

Dear Sir,

Thank you for your valuable comments.

Point 1: The article has been partially improved, some aspects related to my evaluation have been clarified. I understand that in the literature there are data related to the chemical composition of Propolis, but this composition depends on several factors such as plant sources, geographical area, season and obviously other factors.

Response 1:

Thank you for your valuable comments.

  • Our product was purchased from the faculty of Agriculture, Cairo University, and we have no data about the source and what about bees feeding and season of production.

Point 2: How did they determine the safe concentration for Propolis and venom?

Response 2:

  • The concentration post the IC50 value showed 90-100 viability considered safe concentration.

Point 2: My question referred to other issues. In any experiment in which the anti-proliferative effect is evaluated, comparison with a control is imperative.

Response 3:

  • Our control was untreated cells showed 100 % viability so we did not showed it in graphs as no need as long as the viability is not less than 100%

Reviewer 2 Report

The authors have improved the quality of their work, however still some figures should be improved, in particular the 9.

Author Response

Dear Sir,

Thank you for your valuable comments.

It was considered and edited.

Regards,

Mostafa

Reviewer 4 Report

Although the main comments were taken into account in preparation of revised manuscript, still two figures, Figures 1 and 7, have no meaning as there is no numerically defined axis. The author should redraw the Figure 1 and 7 using numerical axes.

Author Response

Dear Sir,

Thank you for your valuable comments.

All comments are considered and edited.

Regards,

Mostafa